# Environmental Drivers of Gross Primary Production and Evapotranspiration at a *Robinia pseudoacacia* L. Restoration Plantation

**Nikos Markos** [1] , **Kalliopi Radoglou** [1] **and Mariangela N. Fotelli** [2,*]

1    Department of Forestry and Management of the Environment and Natural Resources, Democritus University of Thrace, 68200 Orestiada, Greece; nikos.markos@gmail.com (N.M.); kradoglo@fmenr.duth.gr (K.R.)
2    Hellenic Agricultural Organization Dimitra, Forest Research Institute, Vassilika, 57006 Thessaloniki, Greece
*    Correspondence: fotelli@elgo.gr; Tel.: +30-2310-461172 (ext. 238)

**Abstract:** Black locust is the second-most-often planted tree worldwide, particularly for restoration plantations, but drought dieback and growth declines are being reported. Currently, we lack information on these ecosystems' water and carbon fluxes, in relation to climatic variability. Here, we employed eddy covariance to determine the gross primary production (GPP) and evapotranspiration (ET) of a black locust post-mining restoration plantation in NW Greece over c. 2.5 years. Additionally, we applied Generalized Additive Models (GAMs) to study the effects of key environmental drivers (vapour pressure deficit—VPD, soil water content—SWC, solar radiation—Rg and enhanced vegetation index—EVI) on GPP and ET during summer months. Both diurnally and seasonally, GPP increased with increasing Rg, SWC and EVI, but was saturated after certain thresholds (Rg: 400 W m$^{-2}$, SWC: 25%, EVI: 0.65). In contrast, GPP declined strongly with increasing VPD. Overall, GPP was maintained at a high level, at the cost of ET, which constantly raised with increasing solar radiation and SWC and was not responsive to enhanced VPD, indicating a non-conservative water use. At present, these black locust plantations exhibit favourable productivity and no drought stress, but increasing VPD in the context of climate change may, ultimately, negatively impact these ecosystems.

**Keywords:** black locust; eddy covariance; evapotranspiration; gross primary production; vapour pressure deficit; solar radiation; soil water content; EVI; generalized additive models

## 1. Introduction

The eddy covariance method is a powerful tool for evaluating fluxes of carbon dioxide and water between terrestrial ecosystems and the atmosphere over different temporal scales (e.g., [1]). Among other parameters, eddy covariance is used for the estimation of a forest ecosystem's gross primary production (GPP) and evapotranspiration (ET). GPP represents total ecosystem $CO_2$ uptake, while ET is the sum of evaporation from soil, canopy and litter interception, and plants' transpiration. On average, ET is responsible for the return of 70% of precipitation to the atmosphere globally [2], and for the majority of water leaving semi-arid regions [3]. On a global scale, the annual GPP of forest ecosystems is on average approximately $53.7 \pm 4.8$ Pg C yr$^{-1}$ [4]. Both parameters are key variables to understand the energy, water and carbon cycles in forests and to link them with ecosystem functioning and climate feedbacks [5,6]. The coordinated responses of GPP and ET to environmental conditions are vital for ecosystems' performance. While elevated ET may increase ecosystem vulnerability, particularly in drought-prone regions, improved GPP and water use efficiency can enhance the carbon fixation and storage of an ecosystem.

Meteorological and soil water conditions may strongly affect ecosystem water and carbon fluxes through their control of canopy transpiration and evapotranspiration [7–9], as well as ecosystem productivity [10], which is driven by plant photosynthesis. At a

short temporal scale, the optimum equilibrium between maximizing photosynthesis and minimizing water losses through transpiration is governed by stomatal responses to solar radiation and vapor pressure deficit—VPD [8,11]. Net radiation may be a strong determinant of ET and GPP on a monthly basis [12]. In the long-term, ET has been found to follow the variations in precipitation [12,13], but this was not the case for GPP [14]. Moreover, the effect of air temperature is critical for ecosystem evapotranspiration, particularly at sites characterized by warm and dry summers [15]. There, the summer evapotranspiration deficit may negatively impact ecosystem productivity [16].

Drought effects on ET and GPP are also the result of a combination of limited soil water supply and enhanced VPD. Intense and prolonged drought events have the strongest influence on GPP [10], while in extremely dry years soil water may control both the water and carbon dynamics in temperate forests; declining GPP and ET were evident with decreasing soil water, particularly so in broadleaf rather than in conifer forests [17]. However, the effect of soil water availability is complex, being controlled by the root depth of the trees [18]. On the other hand, the exact effect of VPD on both carbon and water fluxes at an ecosystem level is still a topic of debate. Additionally, VPD is strongly coupled with soil water content (SWC), making the distinction of the drought effects on ecosystem fluxes quite difficult. In the context of ongoing climate change and the established increase in VPD [19–22], it is crucial to assess in detail the effect of VPD on ecosystem fluxes.

*Robinia pseudoacacia* L. (black locust) is native to the NE Unites States, and was introduced to Europe in the early 17th century [23]. Nowadays, it also occurs in Asia, southern Africa and Australia and it is the second-most-often planted tree species worldwide [24]. Black locust exhibits great plasticity under diverse climatic conditions, indicating a high acclimation potential for climate change [25]. The species is extensively used for the establishment of plantations for soil and landscape reclamation at degraded sites, due to its advantageous traits. It is a light-demanding, fast-growing and drought-tolerant legume tree [26] that regenerates and spreads rapidly with root-suckers [27]. It also has a high di-nitrogen fixation capacity; its roots develop symbiosis with N-fixing rhizobia and also interact with arbuscular mycorrhizal fungi that grow in the rhizosphere, allowing black locust to cope with environmental stresses such as aridity and low nutrient availability [28]. Based on its favourable properties, and despite its invasiveness [26], black locust has been widely planted for the restoration of areas that are degraded due to anthropogenic activities, such as intensive agriculture and mining in several countries [29–31].

To date, black locust occurs in forty-two European countries, in most of which it is naturalized, and it occupies an area of 2.3 million ha in Europe [24]. *R. pseudoacacia* plantations are established in many semi-arid regions around the world, and growth decline and drought dieback are often experienced (Refs. [32,33] and the literature therein). However, to our knowledge, there is scarce information on the ecophysiological responses of black locust forests or plantations to climatic drivers at the ecosystem level [8], particularly when it comes to data collected by eddy flux towers [9]. The accurate determination of black locust GPP and ET by means of Eddy Covariance will provide the necessary ground measurements for the development, calibration and cross-validation of ecological, biogeochemical and satellite-based models for the prediction of these parameters (e.g., [34–38]). Furthermore, the assessment of black locust ecosystems' water and carbon balance and the in-depth understanding of their covariation with environmental drivers will contribute to our knowledge of the responses of the terrestrial biosphere to climate change, thus improving climate projections [39], and will contribute to the sustainable management of water resources and ecosystems [12], which is crucial in the context of ongoing climate change.

In a previous study, Markos and Radoglou [9] tested the combined effect of environmental variables, such as net radiation, VPD and plant phenology, on the ET of the studied black locust plantations by applying multiple linear regression (MLR) and highlighting the partial significance of each variable. Although significant models were produced, this procedure was applied only to ET and was limited by the small dataset. Additionally, the presumed linearity between the environmental variables and the ET for the MLR ap-

plication could have biased the results. Here, we used a larger dataset of c. 2.5 years to assess the annual GPP and ET and their seasonal variation. Furthermore, we employed Generalised Additive Models (GAM) to isolate and better characterize the effect of different environmental drivers (VPD, SWC, solar radiation, Enhanced Vegetation Index—EVI) on both GPP and ET, during the summer months and within certain thresholds that are important from an ecophysiological point of view. EVI was included in the GAMs as a proxy of the vegetation phenological stage, which is related to ecosystem total leaf area and is, thus, expected to have an impact on ecosystem fluxes. Thus, this study aimed at (a) assessing GPP and ET, and their changes during two consecutive summers, characterized by variability in climatic parameters (particularly VPD and SWC) and (b) understanding the relationships between environmental drivers and ecosystem carbon and water fluxes in the studied *R. pseudoacacia* planted ecosystem.

## 2. Materials and Methods

### 2.1. Study Site

The study site was a *Robinia pseudoacacia* plantation located in Amyntaio, NW Greece (40.59° N, 21.65° E, 700 m a.s.l.; Figure 1), which is part of the restoration plantations established at the Western Macedonia Lignite Centre by the Hellenic Public Power Corporation, which nowadays occupy an area of 2570 ha [31].

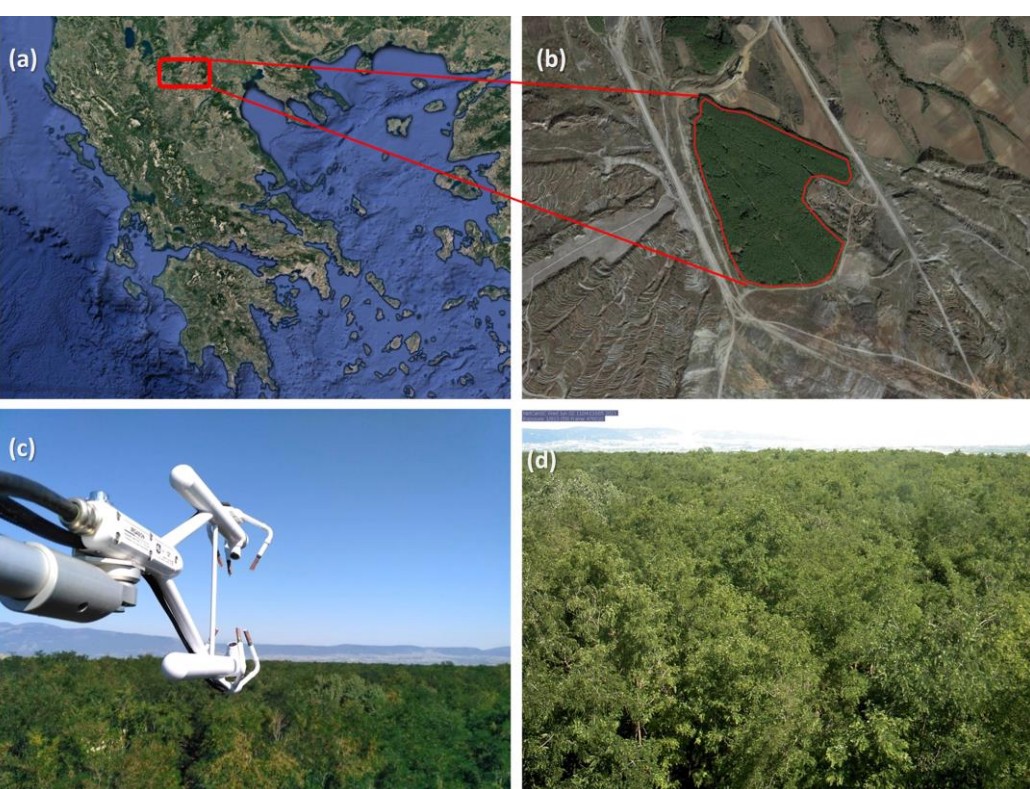

**Figure 1.** Map of the study area and its location in NW Greece (**a**,**b**). Photos of the equipment and the studied plantation from the eddy covariance tower (**c**,**d**).

The plantations are established on open-cast, post-mining depositions resulting from the extraction of lignite and consisting of overburden material. This substrate is generally very poor in nutrients; it has a texture of sandy clay loam (on average 48.4% sand, 28.5% silt, 23.1% clay) and is characterized by alkaline pH (Mean 7.9) and high calcium carbonate content (Mean 27.2%), due to the fact that the mine spoils were created by newly dumped marls [40]. Thus, these plantations aim to withhold erosion, restore the landscape and improve the carbon footprint of the area [31,41].

The studied plantation grows on a flat area and is approximately 20 years old with a mean tree height of 13.5 m. Based on these traits, tree height has peaked [24] and the stand is close to reaching maturity and maximum growth [42–44]. The understorey vegetation consists of perennial grasses, dominated by *Cynodon dactylon* Pers., with a significant contribution to the phenology and fluxes for the period before leaf expansion and after leaf fall [9].

The mean annual precipitation and the mean annual temperature for the period 2010–2020 were $510 \pm 156$ mm and $13.36 \pm 0.92\,^{\circ}$C, while a xerothermic period occurred in July and August (Amyntaio Climate Station of the National Observatory of Athens, http://penteli.meteo.gr/stations/amyntaio/; accessed on 1 April 2023).

### 2.2. Meteorological and Eddy Covariance Measurements

Eddy covariance measurements were performed with an eddy tower installed at the study site in July 2019. The tower is equipped with an open path $CO_2/H_2O$ IRGA analyser with an integrated sonic anemometer (Irgason, Campbell Scientific Inc., Logan, UT, USA), enhanced barometer (PTB 110, Vaisala Inc, Vantaa, Finland) and autonomous air temperature probe. The measurement height is 17.5 m above ground, thus 4 m above mean tree height. The tower is also equipped with a full meteorological station, which includes a photosynthetically active radiation (PAR) sensor (SKP-215, Skye Instruments Ltd., Llandrindod Wells, UK), a pyranometer (SKS 1110, Skye Instruments Ltd., Llandrindod Wells, UK) for monitoring solar radiation (Rg), a temperature/humidity probe (HygroClip2 Advanced, Rotronic Inc., Bassersdorf, Switzerland) for recording air temperature and air relative humidity, a rain gauge (Rain-O-Matic Professional, Pronamic Inc., Skjern, Denmark) and a sensor for SWC measurements at 12 cm depth (CS655; Campbell Scientific Inc., Logan, UT, USA). Eddy data were sampled with a 10 Hz rate and meteorological data were stored with a 30 min time step. VPD was calculated by air temperature and air relative humidity, according to Campbell and Norman [45].

Raw eddy data were processed with the EddyPro 7 software (Licor Biosciences, Lincoln, NE, USA) with default settings, and half-hour fluxes were estimated. Outliers and spikes were detected and removed with the use of the double-differenced time series, using the median of absolute deviation about the median [46]. Gap filling was performed with the marginal distribution sampling (MDS) methodology after u* filtering, and flux partitioning was based on the night-time fluxes method [47]. These post-processing analyses were performed with the use of the REddyProc package [48] available with Rstudio software in R programming language. Full measurements were performed during the period August 2019–December 2021, and were used to assess the annual GPP and ET, as well as their seasonal variation at the study site. In order to determine the effect of environmental parameters on GPP and ET, we focused on the period from June to August (see below "Ecophysiological and statistical analysis").

### 2.3. Assessment of Seasonal Phenological Fluctuation

The study site is characterized by intense phenological differentiations during the growing period, which have a high impact on fluxes [9] due to both the effect of the understory grass layer and the greenness fluctuation of black locust canopies during the vegetation period. The EVI was used as a representative index of the vegetation phenological stage (e.g., [49,50]). For the estimation of EVI we used the products of Sentinel 2 msi (https://sentinel.esa.int/; accessed on 10 November 2022), which offer the advantage of high resolution ($10 \times 10$ m) and frequent acquisition. EVI was estimated for cloud free images for the period 2019–2021, according to Equation (1).

$$\text{EVI} = 2.5 \times (\text{nir} - \text{red})/(\text{nir} + 6 \times \text{red} - 7.5 \times \text{blue} + 1) \tag{1}$$

The study site is located inside overlapping orbits of the satellite, thus the acquisition frequency was very high (a new image was available every 2–3 days). A Loess smooth function, followed by interpolation, was applied to the estimated EVI values in order to estimate the daily values of the index.

*2.4. Ecophysiological and Statistical Analysis*

The ecophysiological analysis focused on the assessment of the impact of the environmental variables on ecosystem fluxes GPP and ET. The effect of environmental parameters on GPP and ET was determined during the period from June to August, when xerothermic conditions are likely to occur, foliage development of black locust is completed and when understorey grass vegetation is wilted, fully decomposed and, thus, does not affect ecosystem fluxes.

As the environmental parameters have simultaneous and added impacts on fluxes, the distinction of the partial impact of each environmental variable is quite difficult under field conditions. For the assessment of the partial impact of each variable on ecosystem fluxes on a daily basis, we applied Generalized Additive Models (GAMs) with the use of the mgcv package in R language [51]. GAMs allow the fitting of non-linear relationships between variables by using a sum of smooth functions of predictors, while maintaining the interpretability of general linear models. GAMs were applied to mean daily GPP and ET separately, using as independent variables the mean Rg, VPD, SWC and EVI, as a proxy of the vegetation phenological stage. Air temperature was not used as an independent factor in the analysis, as it is included in the VPD calculation, and it was also found to have a minimal and non-significant effect on ecosystem fluxes during the preliminary analysis. Each GAM was run as a single model that included all independent variables simultaneously. The partial dependence of GPP or ET on each independent variable was obtained by changing the selected independent variable for its whole range during the study period while keeping the other parameters constant at their mean values. In order to reduce biases, we used the Generalized Cross Validation (GCV) approach, which is used for smoothness selection and is the default fitting procedure in the mgcv package [51]. The λ value, which is a smoothing parameter in GAMs that does not presume the prior shape of the relationship between the variables, was set to 0.6 for each function to avoid overfitting, without further biases to the general shape of the curves. The number of observations for each variable was 215 (i.e., the number of the respective days during the study period) and the significance of the models was tested at least at $p < 0.01$. Data visualization was performed with the use of OriginPro version 2022b (OriginLab Corporation, Northampton, MA, USA).

## 3. Results

VPD was clearly higher and SWC lower in 2021, compared to 2020, and the difference between the two years was significant (Figure S1b,c). The mean and the max values of VPD were 16.4 and 36.1 hPa, respectively, in 2021 and 10.7 and 22.1 hPa, respectively, in 2020. On the other hand, Rg and EVI were similar between these two years (Figures 2 and S1a,d).

The peaks of both GPP and ET were higher in 2021 than in 2020 (Figure 3). However, 2020 was characterized by a longer period of higher mean daily GPP and ET, which finally resulted in significantly higher mean GPP and ET in 2020 vs. 2021 (Figure S1e,f).

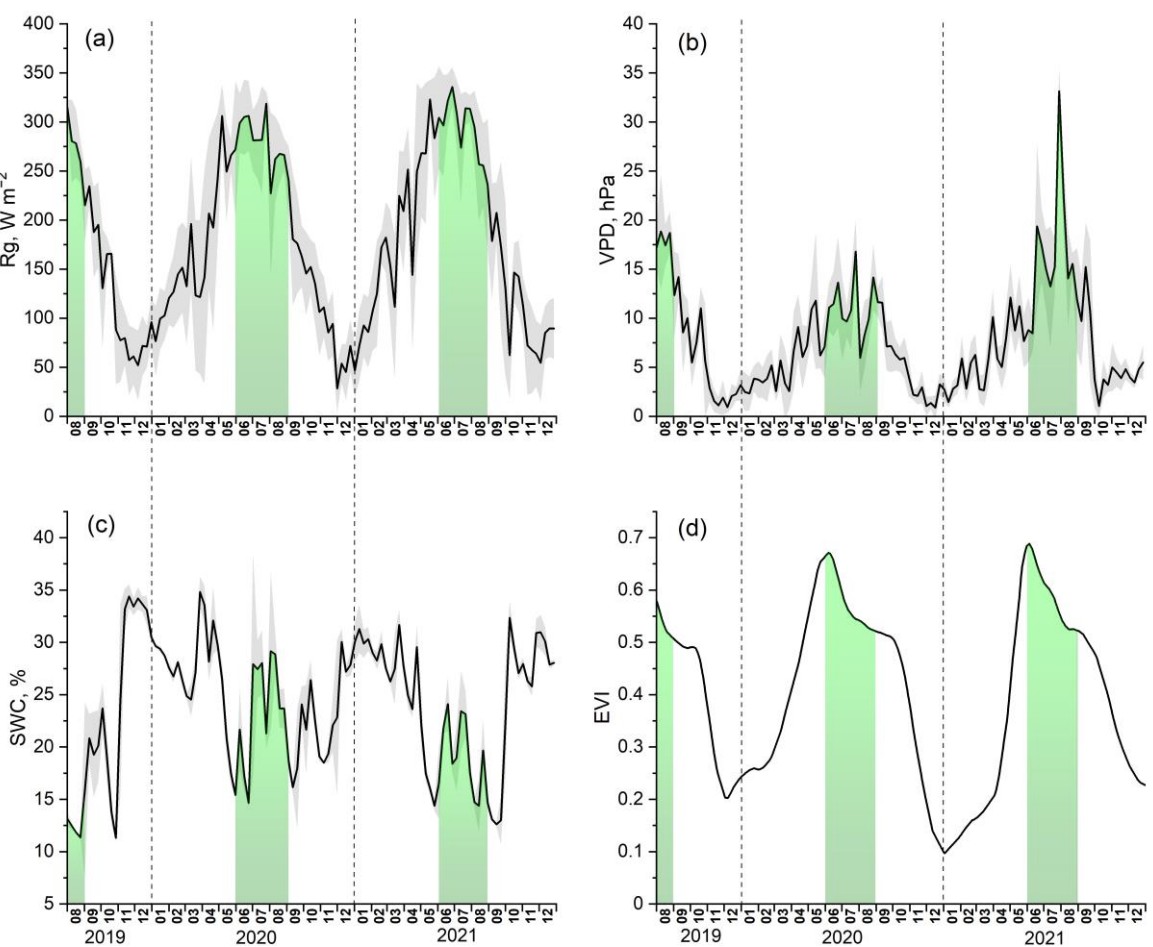

**Figure 2.** Seasonal fluctuations of environmental parameters during the study period: (**a**) Rg, (**b**) VPD, (**c**) SWC, expressed in 8-day averages, and (**d**) EVI, expressed in daily values, after the implication of the Loess smoothing function. The grey bands in plots (**a**–**c**) represent the standard deviation of the means. The green colour marking highlights the summer period (June–August).

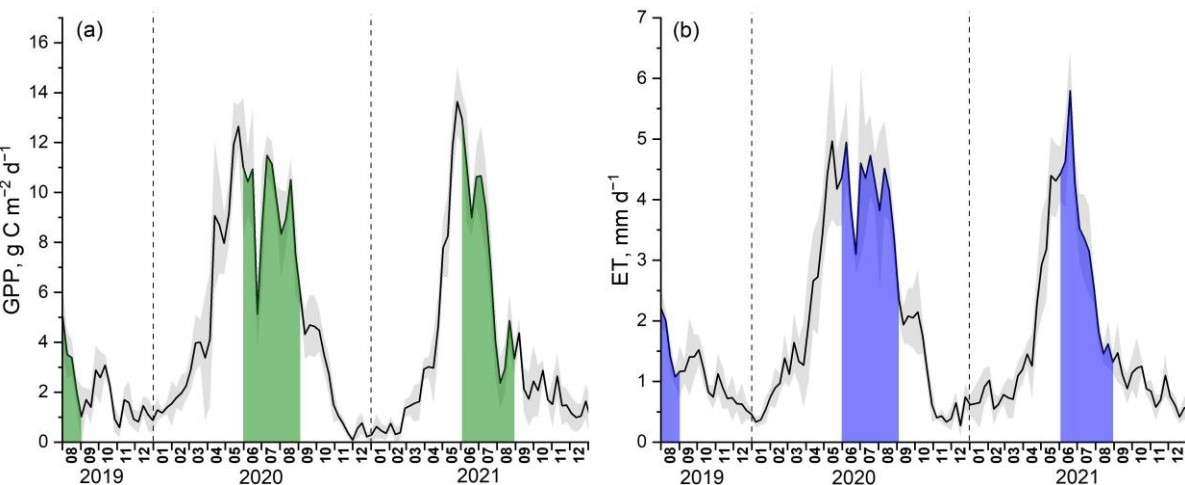

**Figure 3.** Seasonal fluctuation of daily GPP (**a**) and ET (**b**) values for the entire study period, expressed in 8-day averages. The grey bands in plots (**a**,**b**) represent the standard deviation of the means. The green and blue colour marking in plots (**a**) and (**b**), respectively, highlight the summer period (June–August).

GPP peaked earlier in the day (around 09:00 a.m. local wintertime) in comparison to Rg (Figure 4a). This was controlled by VPD, as indicated by Figure 4c; increasing Rg resulted in increased GPP both before and after midday, but GPP was higher when VPD was lower, from the morning until midday. On the other hand, ET closely followed the diurnal variation of Rg (Figure 4b) independent of the diurnal variation of VPD (Figure 4d).

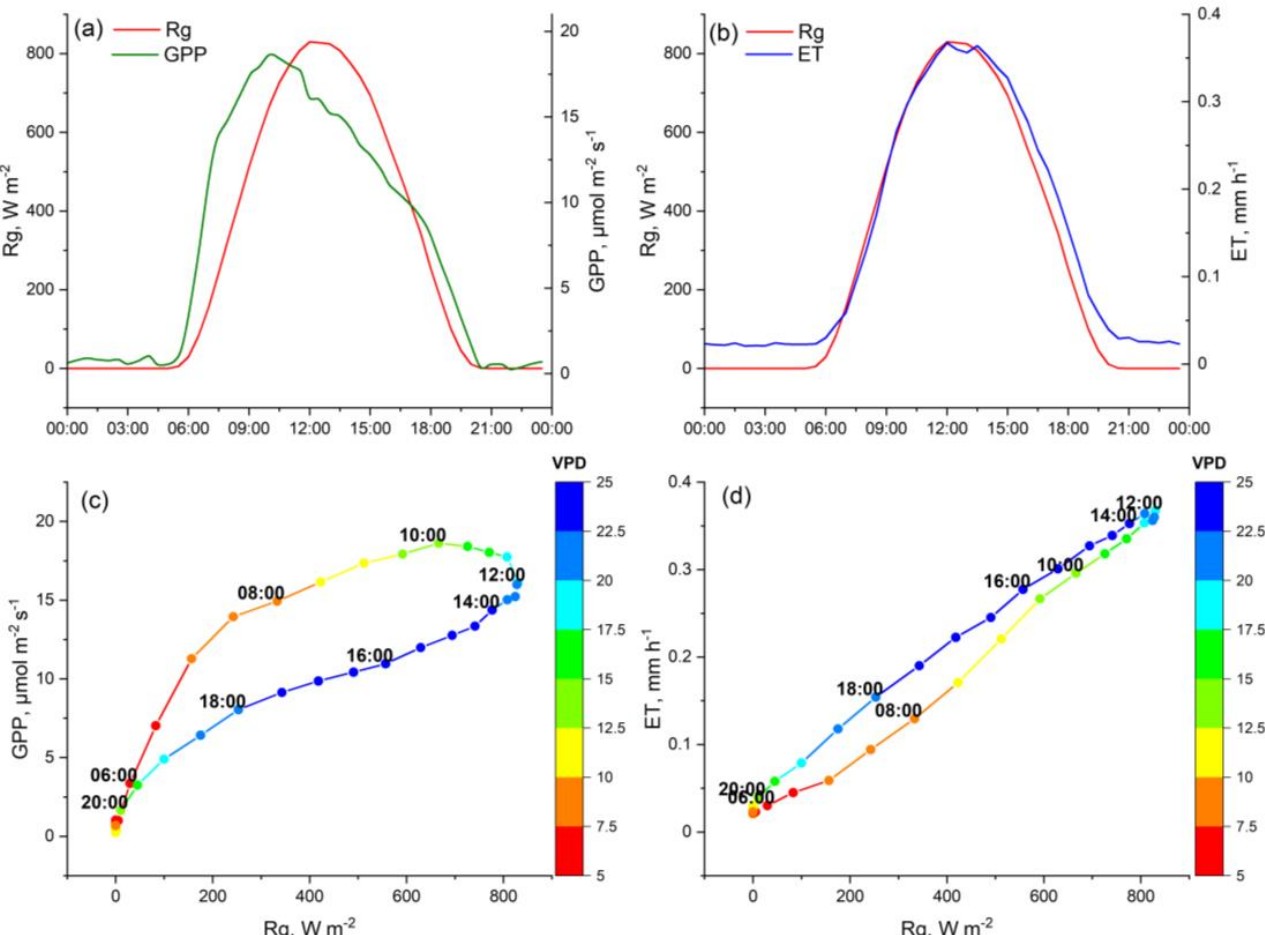

**Figure 4.** Diurnal fluctuation of mean instantaneous GPP (**a**) and ET (**b**) values in relation to the corresponding diurnal Rg patterns. The control of VPD on the effect of Rg on GPP and ET is shown in plots (**c**) and (**d**), respectively. Means of instantaneous values of the summer months (June to August) of the entire study period are presented.

To better illustrate the role of VPD on the relationship of Rg with GPP and ET, these relationships were tested within given VPD thresholds under optimal SWC > 20% (Figure 5). Optimal SWC was chosen based on the relationship between GPP and SWC shown at Figure S2c. For comparability reasons, the same SWC range was used for testing the relationship between ET and Rg (Figure 5b). Under these prerequisites, it was shown that at VPD levels higher than 20 hPa, GPP saturates at Rg around 500 W m$^{-2}$ and it is overall somewhat lower than at VPD below 20 hPa (Figure 5a). On the other hand, no effect of VPD on the relation of ET to Rg was found (Figure 5b).

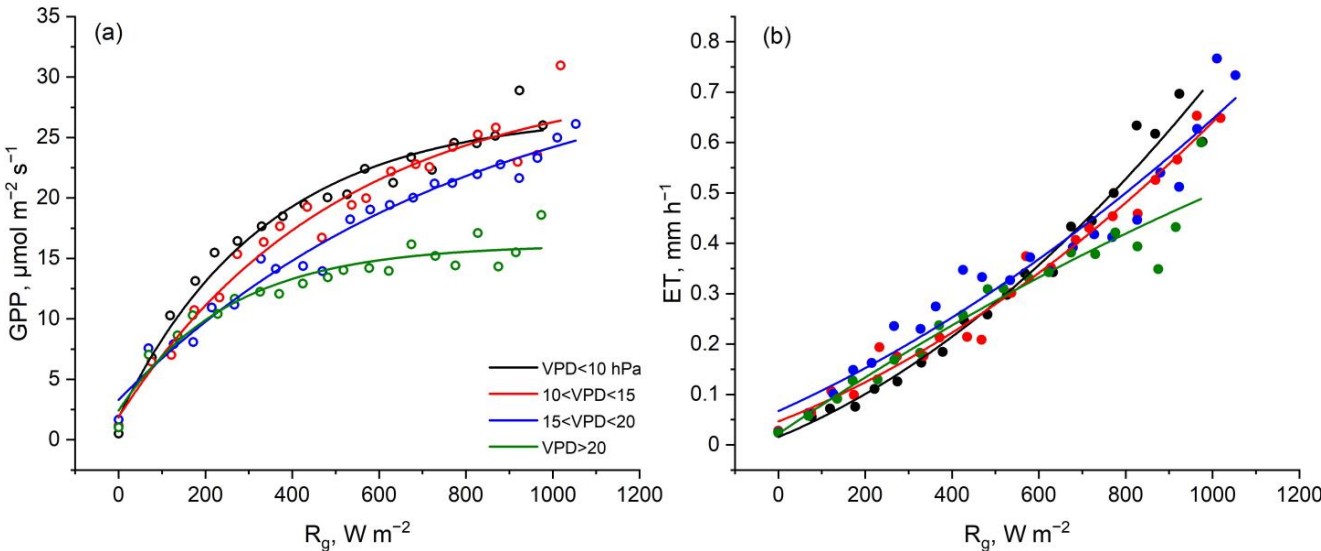

**Figure 5.** GPP and ET dependence on Rg during the period June to August, under optimal SWC conditions (SWC > 20%) and different VPD conditions. The circles (open for GPP and closed for ET) present means of instantaneous values in 50 W m$^{-2}$ Rg intervals for the period June to August during the entire study period (August 2019–December 2021). Black, red, blue and green colour corresponds to VPD < 10 hPa, 10 < VPD < 15 hPa, 15 < VPD < 20 hPa, VPD > 20 hPa, respectively. All regression models are significant at $p < 0.001$. The $R^2$ coefficient ranges from 0.92 to 0.97 for the relationships shown in plot (**a**) and from 0.92 to 0.98 for the relationships shown in plot (**b**).

Instead of applying simple or multiple regression analyses which premise the linearity in the relationships among the independent and the dependent variables, we employed GAM analysis which allows non-linear data to be explained without the assumption of standard-shaped relationships, in order to assess the responses of daily GPP and ET to Rg, VPD, SWC and EVI. All independent parameters were predictors of GPP and ET at $p < 0.001$, except for VPD, which significantly controlled ET at $p = 0.016$ (Table 1). Both GAM models were quite strong, with $R^2$ values at 0.79 and 0.84 for GPP and ET, respectively.

**Table 1.** Outcomes of GAM analysis with simple smoothing functions for each parameter.

| | | GPP | | ET |
|---|---|---|---|---|
| Parameter | F | *p*-Value | F | *p*-Value |
| Rg | 8.726 | <0.001 | 26.27 | <0.001 |
| VPD | 35.744 | <0.001 | 3.51 | 0.016 |
| SWC | 28.561 | <0.001 | 109.11 | <0.001 |
| EVI | 30.053 | <0.001 | 38.71 | <0.001 |
| Regression coefficient | $R^2 = 0.79$ | | $R^2 = 0.84$ | |

GPP increased almost linearly with increasing Rg until the level of c. 400 W m$^{-2}$ (Figure 6a), while the opposite response of GPP to increasing VPD was observed (Figure 6b); GPP declined sharply, as VPD increased above 10 hPa. Additionally, GPP became saturated when SWC and EVI reached 25% (Figure 6c) and 0.65 (Figure 6d), respectively.

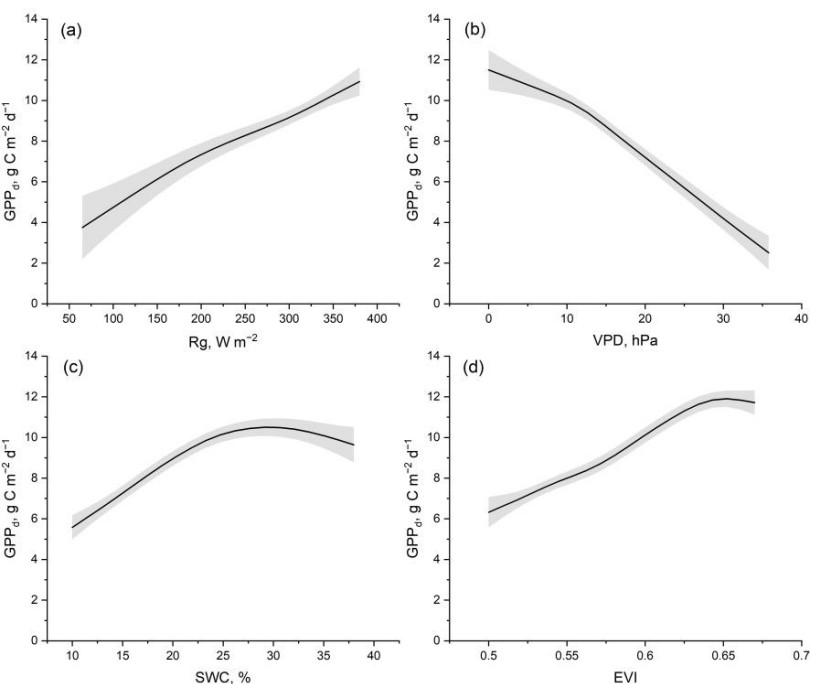

**Figure 6.** Responses of daily GPP to Rg (**a**), VPD (**b**), SWC (**c**) and EVI (**d**), based on GAM analysis. All models are statistically significant ($p < 0.001$). Details of GAM analysis are given in Table 1.

Consistent with GPP, ET also increased almost linearly with increasing Rg (Figure 7a) and the same response of ET was detected in increasing SWC (Figure 7c). However, ET was very little affected by VPD; it presented only a mild decline as VPD decreased to approximately 20 hPa and did not react to any further decrease in VPD (Figure 7b). Almost no response in ET was found up to EVI levels of around 0.58, while ET increased with higher EVI values (Figure 7d).

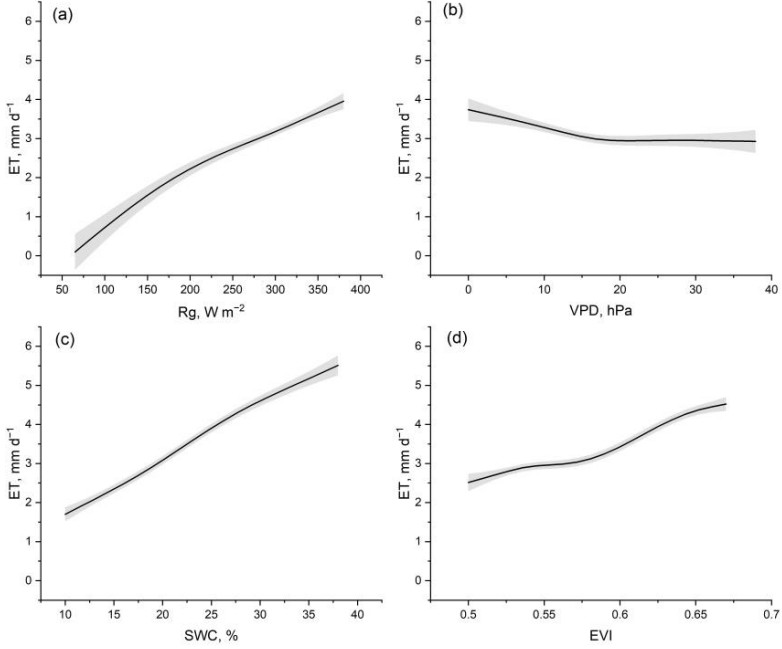

**Figure 7.** Responses of daily ET to Rg (**a**), VPD (**b**), SWC (**c**) and EVI (**d**), based on GAM analysis. ET response to VPD is significant at $p = 0.016$. All other responses are statistically significant at $p < 0.001$. Details of GAM analysis are given in Table 1.

## 4. Discussion

To date, the application of the eddy covariance technique in the *R. pseudoacacia* ecosystems has been limited to the determination of $CH_4$ fluxes [21], and quite recently to the assessment of stand evapotranspiration [9]. Eddy flux measurements have also been performed in a cork oak–black locust mixed ecosystem, where black locust accounted for less than 20% of the forested area [52]. There is, therefore, scarce knowledge about the carbon and water budget of black locust at the ecosystem level, although the species is the second most often used for the establishment of forest plantations globally [24]. To bridge this gap, we established an eddy flux tower in a black locust plantation in NW Greece and assessed the GPP and ET seasonal and diurnal variation for a period of c. 2.5 years.

In many studies, linear or other pre-defined models have been used to describe the single or combined effect of environmental drivers and GPP or ET (e.g., [53–55]). However, GAMs have the advantage of separating the influence of each single factor of those that may drive GPP or ET at the same time, and assessing the exact form of the relationship between the dependent and independent variables, without having to assume the form of their relationship (e.g., [56,57]). This allowed us to assess the exact influence of the tested environmental drivers (Rg, VPD, SWC, EVI) on GPP and ET and to identify certain thresholds that are important for understanding the ecophysiological responses of black locust ecosystems to these variables.

### 4.1. The Studied Black Locust Plantation Is Characterized by High GPP and ET

Given that no data have been published on the GPP and ET of other black locust ecosystems, comparisons can only be made with other forest ecosystems that prevail in the Mediterranean area or are characterized by a Mediterranean climate. Overall, the annual GPP in the studied plantation was 1914.7 and 1445.0 g C m$^{-2}$ yr$^{-1}$ in 2020 and 2021, respectively (Table S1), and it was similar to the mean annual GPP of 11 other deciduous broadleaf forest sites in Europe within the Fluxnet network (on average 1644.4 $\pm$ 245.12 g C m$^{-2}$ yr$^{-1}$; [58]. However, the GPP of black locust plantations was higher compared to that, either measured by an eddy tower or estimated by models, of other Mediterranean-type ecosystems or ecosystems with a Mediterranean climate, dominated by *Quercus ilex* L. [59] and *Pinus brutia* [37], as well as Chestnut, European beech and other forest species [36]. Furthermore, the GPP of the studied black locust peaked at higher levels (approximately 12–14 gC m$^{-2}$ d$^{-1}$) compared to the respective maximum values recorded in a deciduous oak ecosystem with a Mediterranean climate during a 12 yr period (c. 8 gC m$^{-2}$ d$^{-1}$; [60]. Although these GPP estimations generally refer to mature ecosystems, the differences in GPP may also, to some extent, be due to age effects. Nevertheless, the elevated GPP rates of black locust are in line with the species' high growth rate, particularly at a young age [24,44], even though the studied plantation grows on particularly infertile post-mining depositions [31]. However, an important contributor to the ecosystem's GPP, apart from black locust, is the understorey grasses vegetation which performs a considerable $CO_2$ fixation in the months prior to the initiation of leaf expansion in black locust (January to March), and its contribution to GPP continues until the dry summer period, when it wilts and is quickly decomposed, as similarly observed by others [34]. To assess the seasonal dynamics of the grasses understorey and estimate its GPP, light use efficiency models with phenological indices were developed in an Australian savanna ecosystem [61], but this approach has not been applied at the study site yet.

The annual ET at the studied site ranged from 578.5 to 739.6 mm yr$^{-1}$ (Table S1) and it was comparable to that of 11 Fluxnet broadleaf deciduous European forests (622.7 $\pm$ 133.2; [58]), mixed forests with Mediterranean climate in California, USA [62], and with holm oak and Aleppo pine ecosystems with similar precipitation in Spain [63,64]. With regard to the seasonal fluctuation of ET, Helman et al. [65] recorded the ET minima and maxima of a Mediterranean evergreen coniferous forest in winter and summer months, respectively, in line with the seasonal ET pattern we observed (Figure 3b). In two forest ecosystems with a Mediterranean climate and similar precipitation and mean annual temperature as in our

site, ET peaked earlier than in the present study, in June, and decreased in the following summer months [62]. Similarly, tree transpiration and the water requirements (transpiration and vegetation and soil evaporation) in black locust ecosystems were maximum around June [33,66]. On the other hand, in the studied black locust ecosystem, ET was maintained at high levels until July in both study years and even peaked during the xerothermic period in 2020 (Figure 3b), indicating a non-conservative water use. In the same context, it was observed that the ET of the tree layer of a mixed *Olea sylvestris–Quercus suber* Mediterranean ecosystem was not greatly affected by summer drought and that the wilting of the grasses understorey in the summer had an influence on ecosystem ET [34]. Although grasses' wilting and replacement by bare ground was evident in the xerothermic period in the studied plantation, we were not able to distinguish its effect on evapotranspiration at the ecosystem level. Nevertheless, even if such an effect cannot be excluded, ET was still high overall, particularly during 2020.

### 4.2. Physiological and Environmental Controls That Contribute to Elevated GPP and ET

The achievement of high GPP and ET during the summer period, which indicates the resilience of black locust to xerothermic conditions, may be partially supported by the strategy of paraheliotropism, the movement of leaves so that the leaf lamina is oriented obliquely to the sun direct rays [67]. Such a mechanism is adopted by the leaves of black locust, and it allows the protection of the photosynthetic apparatus against high irradiance and temperature, thus enabling $CO_2$ fixation and transpiration in warm and dry summer days [68].

GPP was also favoured by the increasing Rg during the summer and when soil water availability was adequate, but it saturated at SWC levels above 25% (Figure 6a,c). Under such water availability, Mediterranean-type ecosystems experience an optimal water status, illustrated by the absence of sensitivity to precipitation [60]. Increasing vegetation greenness during the summer, as expressed by EVI, also positively influenced GPP (Figure 6b), as similarly observed in other ecosystems with a Mediterranean climate ([60,65]). In addition, the observed optimum GPP was probably supported by the time lag between GPP and Rg or VPD; we recorded an early achievement of maximum productivity at around 09:00 a.m., before the occurrence of high Rg and VPD (Figure 4a,c). GPP generally declines with increasing VPD, as has been evidenced across different ecosystems ([21,69]), and it is attributed to photosynthetic limitation due to decreased stomatal conductance (e.g., [70]). In this context, increasing VPD after midday controlled GPP, in line with [52], and resulted in its c. 50% decrease at VPD above 20 hPa (Figure 5a). Moreover, the elevated VPD after midday caused a hysteresis which did not allow GPP to recover to morning levels, as similarly observed in a Mediterranean Eucalypt site [53]. Nevertheless, although it declined, GPP was still considerable in the afternoon (Figure 4a), thus contributing to the overall high daily and annual GPP of the ecosystem.

In line with the responses of GPP, ET was positively affected by Rg, EVI and SWC, but without being saturated at any threshold of these parameters (Figure 7a,c,d). On the contrary, no influence of VPD on ET was detected (Figures 4d, 5b and 7b). Plant transpiration increases with increasing VPD up to a certain threshold [22], while the ET response to elevating VPD may either be negative or positive, indicating a conservative or consuming water strategy, respectively [71]. However, ET was almost non-responsive to VPD in our study site, indicating the absence of stomatal regulation to corresponding vapor pressure deficit during the summer. This contradicts previous studies on the leaf level, which indicated that black locust exhibits an isohydric stomatal control and avoids an increase in transpiration due to summer evaporative demand and decreasing soil moisture [15,33]. Thus, the increase in ET in mid-summer, independent of the elevated VPD, is probably associated with an, on average, adequate water supply. In the same context, it is reported that at high water availability black locust did not impose stomatal control on transpiration even at VPD exceeding the threshold of 20 KPa [72], above which GPP was impacted in our study (Figure 5a).



However, as VPD continuously rises with the ongoing climate change [19–22], it cannot be excluded that both the GPP and ET of black locust plantations may be impacted in the near future.

## 5. Conclusions

During the summer period, GPP responded positively to increasing solar radiation, EVI and SWC, but strongly declined with elevating VPD, particularly after midday. Despite the negative effect of VPD on GPP, the black locust plantations maintained overall high GPP levels; higher than the values reported in many other forest ecosystems with a Mediterranean climate, verifying the high productivity and growth rate of the species. This was achieved at the cost of water use, as ET constantly increased with increasing solar radiation and SWC independent of the rising VPD, indicating that black locust did not experience serious water depletion even during the summer. Thus, the studied plantations in NW Greece do not currently undergo any growth decline and drought dieback, as evidenced in several *R. pseudoacacia* restoration plantations established on semiarid regions around the world, despite the fact that they grow under adverse conditions on post-mining depositions. Future research should focus on assessing the carbon assimilation and evapotranspiration of the understorey of such black locust plantations, which have a strong seasonal fluctuation and different phenology than black locust, in order to partition its contribution to overall ecosystem GPP and ET.

**Supplementary Materials:** The following supporting information can be downloaded at: https://www.mdpi.com/article/10.3390/f14050916/s1, Figure S1: Box charts presenting the population statistics of the variables Rg (a), VPD (b), SWC (c), EVI (d), GPP (e) and ET (f) for the years 2020 and 2021, Figure S2: Responses of GPP to solar radiation (a), VPD (b), SWC (c) and EVI (d), Figure S3: Responses of ET to solar radiation (a), VPD (b), SWC (c) and EVI (d), Table S1. Total evapotranspiration (ET) and gross primary production (GPP) in the studied black locust plantation, during the entire year and during the summer period (June to August) in the years 2020 and 2021.

**Author Contributions:** Conceptualization, N.M. and K.R.; methodology, N.M.; formal analysis, N.M.; writing—original draft preparation, N.M. and M.N.F.; writing—review and editing, N.M., M.N.F. and K.R.; visualization, N.M.; project administration, K.R.; funding acquisition, K.R. All authors have read and agreed to the published version of the manuscript.

**Funding:** This research has been co-financed by the European Union and Greek national funds through the Operational Program Competitiveness, Entrepreneurship, and Innovation, under the call RESEARCH–CREATE–INNOVATE (project: COFORMIT. T1EDK-02521).

**Data Availability Statement:** The data presented in this study are available on request from the corresponding author.

**Acknowledgments:** The authors acknowledge the Hellenic PPC (Public Power Corporation) S.A. for the provision of assisting personnel and necessary infrastructure during field campaigns and data collection. Special thanks are due to Lamprini Patmanidou, Simela Andreadi and Tryfon Barbas, and their teams.

**Conflicts of Interest:** The authors declare no conflict of interest.

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
