# Peer review of "Environmental Drivers of Gross Primary Production and Evapotranspiration at a Robinia pseudoacacia L. Restoration Plantation"

_forests, doi:10.3390/f14050916_

Round 1

Reviewer 1 Report

The article is very interesting and should be published in Forests, but there are some major comments.

The Materials or a Methods section needs to be expanded in the main text. “2.1. Study site”, “2.2. Meteorological and eddy covariance measurem” and “2.3. Phenology?” are described very well. But “2.4. Ecophysiological analysis” is too general and simple. Please describe the definition, data source, acquisition method, calculation formula, quantitative method, grouping design, visual processing, repetition rate, etc. of the key indexes, including Generalized Additive Models (GAMs), gross primary production (GPP), evapotranspiration (ET), vapor 16 pressure deficit - VPD, soil water content – SWC, solar radiation - Rg in detail,in digital and in individual.

“2.3. Phenology” a Material or a Method? may be phonological evaluation? or EVI estimation?

Statement about “statistics analysis” needs to be supplied.p-value” “R2

What’s the topic of the Discussion. Some or few key points are hard to be get by the readers?The Conclusion in this paper should be more concise, e.g. just the novel finding and / or the most important conclusion, preferably no references, less prognosis.

One more question. Together with “the period August 2019 – December 2021” and “only the summer months (June, July and August)”, only “August” showed three repetitions (2019, 2020, 2021), but not June and July (2020, 2021). Now, it is 2023. How about the data of June, July and August 2022. Why not?

Author Response

Dear Reviewer,

thank you for your comments. Please see the attachment for our point-by-point response.

Reviewer 2 Report

This study determined the GPP and ET during 2.5 years of Robinia pseudoacacia L. restoration plantations by eddy covariance. The influence of key environmental factors on restoration plantations such as VPD, SWC, Rg and EVI in the summer was studied by GAMs. The authors suggest that an increase in VPD may eventually have a negative impact on R. pseudoacacia restoration plantation ecosystems in a changing climate. This study demonstrates the covariation between R. pseudoacacia plantation ecosystem and environmental factors, which is helpful to promote the management and development of plantation ecosystem. In addition, the paper is rich in content, reasonable in structure and accurate in expression.

Introduction: is well written.

Line 107: As you mentioned there is a considerable greenness fluctuation during the year, why you chose only summer months (June, July and August) rather than other months such as April and May?

Results:

Line 198: Try not to write sentence like "Figure 2 presents……”. References to diagrams should be enclosed in parentheses. The same problem occurs in Line 213, Line 223, Line 226 and Line 275.

Using the first letter to indicate months is not intuitive in Figure 2 and Figure 3. I suggest replacing it with numbers.

Author Response

(The authors gave the same response as above.)

Reviewer 3 Report

The authors presented an interesting study that sheds light on the drivers of GPP and ET in a young black locust plantation. The study deserves publication provided certain improvements are introduced in the document. Some remarks are presented hereafter:

- Lines 63-64: In the following text,“In the context of the ongoing climate change and the established increase in VPD the assessment of its accurate effect of ecosystem fluxes is trivial.”, What do you mean? Also in lines 93-94. What do you mean by trivial?

-In line 101: "...we use a larger dataset of c.2.5 years..." whereas in fact "...we used only data that refer to the period of full leaf expansion, i.e. only the summer months (June, July and August) during these years". Misunderstandings in the study description should be avoided; when reading that you analyzed "a larger dataset of c.2.5 years" one can expect 30 months of data. In other parts of the document it is mentioned that the study period goes from august 2019 to december 2021. Apparently, the full dataset was used only to prepare some plots; the actual analysis was done only on the summer months. In addition to the need of express more clearly the actual studied period, it is not explained why did the authors took the decision of discarting other months (the data were available) which can also be relevant to understand the connection between the fluxes and the drivers (the spring, for instance). More clarity on this point is needed.

- Apparently, the analysis did not account for time delays in the connections between environmental drivers and GPP/ET. Is it right? Perhaps it should be mentioned as well.

- Line 183-184: perhaps the Methods section is the right place to elaborate more on why did you choose for the GAM analysis and include a brief word on what the GAM analysis is. Likewise, the use of GCV: please explain more on this method/technique. What is the method about? What are the main principles? why did you choose it. Cite relevant reference(s).

- Lines 308-311: "...the GPP of black locust plantations was higher compared to that....and other species" Perhaps the age of the plantation is the key issue here (?).

Line 101: “...we use a larger dataset of c.2.5 years…” where as in fact (line163-164) “… we used only data that refer to the period of full leaf expansion, i.e. only the summer months (June, July and August) during these years”.

Line 101: “...we use a larger dataset of c.2.5 years…” where as in fact (line163-164) “… we used only data that refer to the period of full leaf expansion, i.e. only the summer months (June, July and August) during these years”.

Author Response

(The authors gave the same response as above.)

Round 2

Reviewer 1 Report

Thank the authors about their effort to improve the manuscript. The authors have comprehensively revised this paper. It should be published after some minor revisions.

1. The citation format of references needs to be unified. such as Line 31, 98, 165, 180, 358, 419. please check the whole paper.

2. The abbreviation format also needs to be consistent. “vapor pressure deficit – VPD” in line 49, 160, 208, 414. “solar radiation” in line 17, 21, 107, 154, 429, 435. “soil water content” in line 107, 116, 157, 208. The full name of the abbreviation is only given when it first appears. please check the whole paper.

3. If multiple references are cited in one sentence, preferably play all of them at the end of the sentence, or split into two or more sentences. such as Line 50, 51 “On the long-term, ET was found to 50 follow the variations in precipitation [13], [12], but this was not the case for GPP [14].”, line 399-401 “GPP generally declines with increasing VPD, as it has been evi-399 denced across different ecosystems [67], [21] and it is attributed to photosynthetic limita-400 tion due to decreased stomatal conductance (e.g. [68]).” please check the whole paper.

4. Preferably add spaces before and after the symbol. “p<0.001” to “p < 0.001”. “p=0.016” to “p = 0.016”. Equation 1 “EVI=2.5 (nir-red)/(nir+6*red-7.5*blue+1)”

Author Response

Dear Reviewer 1,

thank you for your comments. Please find our point-by-point response in the attached document. We think all issues were addressed and we hope the paper can be accepted for publication at its current form.
